# Soft tissue lesion detection in mammography using deep neural networks for object detection

**Jonas Teuwen***
Radboud University Medical Center, Nijmegen, the Netherlands
jonas.teuwen@radboudumc.nl

**Sil C. van de Leemput**
Radboud University Medical Center, Nijmegen, the Netherlands
sil.vandeleemput@radboudumc.nl

**Albert Gubern-Mérida**
Screenpoint Medical, Nijmegen, the Netherlands
albert.gubernmerida@screenpointmed.com

**Alejandro Rodriguez-Ruiz**
Radboud University Medical Center, Nijmegen, the Netherlands
alejandro.rodriguezruiz@radboudumc.nl

**Ritse Mann**
Radboud University Medical Center, Nijmegen, the Netherlands
ritse.mann@radboudumc.nl

**Babak Ehteshami Bejnordi**
Radboud University Medical Center, Nijmegen, the Netherlands
behtesha@qti.qualcomm.com

## Abstract

Computer-aided detection or decision support systems aim to improve breast cancer screening programs by helping radiologists to evaluate digital mammography (DM) exams. Commonly, such systems proceed in two steps: selection of candidate regions, and subsequent false positive reduction of the candidates as either suspicious lesions or inconspicuous breast tissue. In this study, we present a method based on deep learning for automatic detection of soft tissue lesions in DM using a one-step approach. A database of DM exams (mostly bilateral and two views) was collected from our institutional archive. In total, 7192 DM exams (23405 DM images) were acquired with systems from three different vendors (General Electric, Siemens, Hologic), of which 2883 contained malignant lesions verified with histopathology. The performance of our automated detection system was assessed using the free receiver operating characteristic (FROC) analysis. A maximum sensitivity of 0.97 at 3.56 false positives (FP) per image was achieved. The best model achieved a sensitivity of 0.73, 0.45, 0.31 at 0.1, 0.02 and 0.01 FP per image, respectively.

---

*Jonas Teuwen is also with the Department of Imaging Physics, Optics Research Group, Delft University of Technology, Delft, the Netherlands. Babak Ehteshami Bejnordi currently works for Qualcomm Research, Amsterdam, The Netherlands.

1st Conference on Medical Imaging with Deep Learning (MIDL 2018), Amsterdam, The Netherlands.

Overall, the results of our evaluation suggests that our soft tissue lesion detection system can replace current two stage detectors.

# 1 Introduction

Population-based screening programs with mammography are widely accepted as the most efficient way to reduce breast cancer related mortality [18]. Nevertheless, they still have room for improvement. Several studies have shown that a significant number of cancers diagnosed between screening rounds were already visible on previous screening mammograms, where they were wrongly marked as negative [1, 2, 5, 6, 15]. Additionally, it has been demonstrated that there is a significant variability in performance between screening readers, and, therefore, combining assessments by two or more readers improves screening performance [4, 7, 19, 22, 31]. These pitfalls, combined with the increasing scarcity in radiologists, including those specialized in breast imaging, [29] suggest that computer systems not only have the potential to improve breast cancer screening programs, but may hold the key to the subsistence and development of screening programs [3].

Since the first FDA-approved computer-aided detection (CAD) system for mammography in the late 90s, CAD has been widely used in screening, especially in the US where single reading of exams is a more common practice. However, several studies have shown that neither the radiologists performance nor the cost-effectiveness of the programs improve when using CAD, mainly because of the low specificity of these traditional systems [10, 11, 22]. Nevertheless, the recent developments in artificial intelligence techniques for perception tasks, in particular deep neural networks, have greatly improved the performance of such algorithms in many fields of medical imaging [9, 24]. It can, therefore, be expected that a new generation of computer-aided detection or diagnosis systems for digital mammography (DM) might finally yield a significant improvement in breast cancer screening programs.

As a consequence, much attention is being paid to developing deep learning-based CAD systems for DM. Generally, CAD systems proceed in two steps. In the first step, the whole mammogram is processed and regions of interest are selected, the so-called candidate detector. The primary goal of this step is to greatly reduce the number of search locations while achieving a sensitivity near $100\%$. In the second step, the goal is to remove the false positives, while keeping the true positives. Up to now, few reported candidate detectors are based on deep learning technology [8]. The first study to use deep learning for the second stage was performed by Kooi et al. [20], whose neural network was proven to be not significantly different than certified screening radiologists for breast cancer detection on a patch level. In this work, we studied the use of the recently proposed object detection deep learning networks, to combine both the candidate detector and the classification step into one single model.

## 1.1 Related work on object detection

Object detection is the task of finding different objects in an image and classifying them. R-CNN (Regional CNN) [14] and its descendants fast R-CNN [13], and faster R-CNN [28] are among the most popular models for object detection. Ross Girshick et al. [14] proposed R-CNN for accurately identifying objects in an image via bounding boxes. This model first produces a collection of bounding box proposals (e.g. using Selective Search) and stores the ones which overlap a ground-truth object with an intersection over union (IoU; area of overlap divided by area of union) bigger than a predefined threshold. Next, the non-maximum suppression algorithm is used to discard overlapping proposals that have an IoU larger than some predefined threshold with a proposal that has a higher score. Following the creation of proposals, the bounding boxes are cropped and scaled to a standard size and passed to an AlexNet-like [21] network. The features from the last layer of this network are then fed into a support vector machine (SVM) that classifies whether the warped image in the bounding box is an object, and if so what type. Finally, the bounding box proposals are tightened by training a linear regression model to output tighter bounding box coordinates. This linear model is trained on CNN features for the warped image and its relative bounding box coordinates. While R-CNN performs very well for object detection, it is very slow for two main reasons. Firstly, every single region proposal requires a forward pass of the CNN leading to unnecessary redundant computations. Secondly, three different models need to be trained separately which makes the pipeline hard to train.

Fast R-CNN [13] extends this architecture by attending to Regions of Interest (ROIs) directly on the feature maps generated by the CNN for the entire image using *RoIpooling* method, leading to significant speed up of both training and testing and more accurate results. Using this method, the bounding box proposals from the input image are directly projected to corresponding ROIs on the last feature map of the CNN. The ROIs of nonuniform sizes are then pooled (usually using max pooling) to obtain fixed-sized feature maps.

A major bottleneck for the speed of R-CNN and fast R-CNN was the region proposer that is based on selective search. Faster R-CNN [28] extends fast R-CNN by learning the attention mechanism using a region proposal network (RPN). RPN takes as input a set of fixed-size rectangles called anchors that are defined over the last convolutional feature map in a sliding window fashion. RPN generates objectness scores for the anchors and predicts four correction coordinates to move and resize the anchor to the right position. RPN is a more efficient and accurate region proposal than selective search method used in R-CNN and fast R-CNN as it allows to backpropagate the error signal to improve the proposals.

Kaiming He et al. [16] proposed a general framework called mask R-CNN for simultaneously detecting and segmenting object instances in an image. This method extends faster R-CNN by adding a branch which is a fully convolutional network for predicting an object mask in parallel with the existing branch for bounding box recognition. Additionally, mask R-CNN improved the *RoIpooling* method in faster R-CNN by a method called *RoIAlign*. The regions selected by *RoIpooling* in the feature map are usually slightly missaligned from the regions of the original image due to performing coarse spatial quantization. *RoIAlign* is a quantization free layer that preserves the spatial locations by computing the value of each sampling point using bilinear interpolation from the nearby grid points in the featuremap. Even without the mask branch, when training faster R-CNN with the *RoIAlign* method, an improvement in detection performance was observed.

## 2 Materials and Methods

### 2.1 Patient population

This study was conducted with anonymized data retrospectively collected from our institutional archive. The study was approved by the regional ethics board after summary review, with waiver of a full review and informed consent.

Between 2000 and 2016, DM exams from women who attended the national screening program at our collaborator institution, and our institution for diagnostic purposes were included.

All cases with biopsy-proven malignant soft tissue lesions were collected, while normal exams were selected if they had at least two years of negative follow-up. This yielded a total of 7192 DM exams, from which 2316 exams (32%) contained a total of 3023 biopsy-verified malignant lesions. Most exams were bilateral and included two views (cranio-caudal -CC- and medio-lateral oblique -MLO-), resulting in a total of 23,405 images. The exact distribution is summarized in Table 1.

Table 1: Distribution of the digital mammography (DM) exams included in this study.

|  | Total | GE | Siemens | Hologic |
|---|---|---|---|---|
| Unique patients | 6379 | 1663 | 1450 | 3266 |
| DM studies | 7192 | 2246 | 1517 | 3429 |
| normal studies | 4876 (68%) | 1608 (46%) | 1335 (88%) | 2516 (73%) |
| malignant studies | 2316 (32%) | 1221 (54%) | 182 (12%) | 913 (27%) |

### 2.2 Image acquisition and preprocessing

The images from our institutional archive were acquired by four DM machines from three different vendors (Senographe 2000D and Senographe DS, General Electric, USA; Mammomat Inspiration and Mammomat Novation DR, Siemens, Germany; Selenia Dimensions, Hologic, USA;).

Images were preprocessed in four steps. In the first step, an energy band normalization technique [26] was applied to homogenize the contrasts across different vendors. In the second step, all the

mammograms (originally acquired with resolutions ranging from 70 to 100 $\mu$m) were resampled to 200$\mu$m. This resolution is considered a good trade-off between accuracy, memory usage and speed for the detection of soft tissue lesions. Prior to resampling, a Gaussian filter was applied to reduce the associated aliasing effects. In the third step, we cropped the image to the bounding box of the breast with an additional margin of 5 mm, to reduce computational costs. Finally, the intensity of each image was rescaled to the integer range $[0, 255]$, while maintaining the relative original window level.

Transfer learning using supervised features has been successful in several computer vision and medical imaging applications [12, 27]. To take advantage of the computational and potential performance gain by using pretrained networks trained on natural images such as ImageNet [30], we converted our images to RGB by duplicating the gray values over the three color channels. Examples of preprocessed images with annotated lesions are shown in Figure 1.

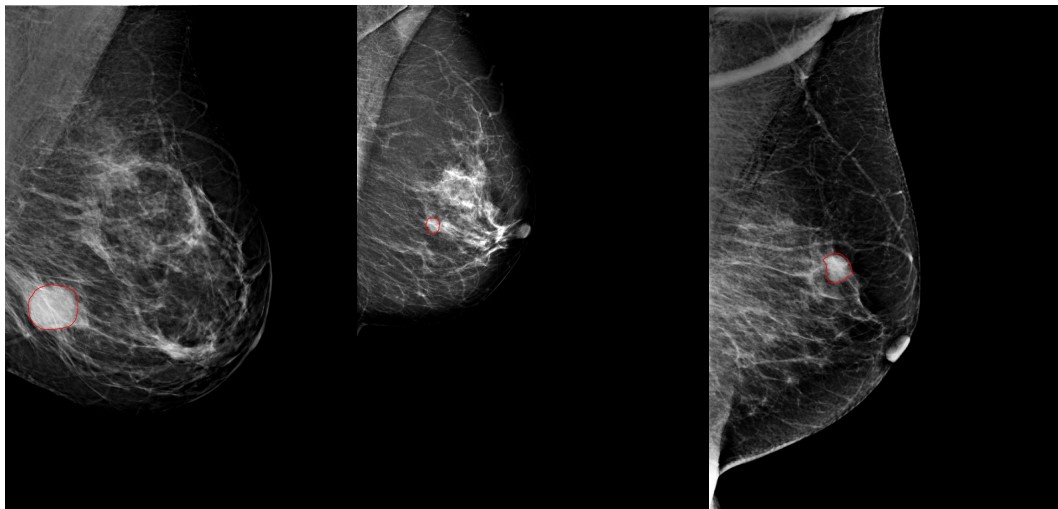

Figure 1: Examples of digital mammography images acquired with machines from three vendors: General Electric (left), Siemens (middle) and Hologic (right); after preprocessing. Biopsy-proven lesions are outlined.

## 2.3  Deep learning network and training protocol

Data was split on a DM study level to avoid bias. Our models were trained on $60\%$ of the dataset and evaluated on the remaining $40\%$.

We evaluated the potential of faster R-CNN and mask R-CNN models with several CNNs as backends. We chose the ResNet-101 and ResNeXt-101 and ResNext-152 [32] (with different bottleneck transformations) architectures as the backend models. The weights for all these networks were initialized using pretrained models on the ImageNet [30] dataset.

All networks were trained on the full image size to provide enough context to discriminate soft tissue lesions. Images of a malignant study were only included when they had accompanying annotations to prevent that incomplete annotations are erroneously misclassified. All models were trained with stochastic gradient descent (SGD) with momentum of 0.9 and a weight decay of 0.0001.

We applied the same learning rate (LR) schedule to all models. We used a steps with decay scheme where we have three different learning rates at equally spaced intervals. The current learning rate was always computed as (current LR = initial LR $\cdot \gamma^{n_{\text{step}}}$). We stopped training after a fixed number of epochs. We take steps at $[0, 30000, 40000]$ iterations for the faster R-CNN models and stop at 45000 iterations, and for the mask R-CNN we divided the number of iterations by two due to time constraints.

At the start of training we used an LR warm-up schedule which started at $^{1}/3^{\text{th}}$ of the initial LR and subsequently linearly increased for the first 500 iterations to reach the initial LR.

In contrast to the original faster R-CNN paper, we used the *RoiAlign* layer instead of the *RoiPool* layer, as the ablation experiments in [16] showed significant improvements in localization using the *RoiPool* layer.

All the convolutional layers of our networks were pretrained on ImageNet. We adapted regional proposal networks in faster R-CNN and mask R-CNN by replacing the single-scale feature map from ResNet-101, ResNeXt-101, and ResNeXt-152 models with feature pyramid networks [23]. To augment the dataset we apply horizontal flipping to the images and ground truths. For the deepest network, the ResNeXt-152, we have additionally applied the same augmentation at test time.

## 2.4   Empirical evaluation

The performance of the model was evaluated using free receiver operating characteristic (FROC) analysis. The FROC curve is defined as the plot of sensitivity versus the average number of false positives per scan [25].

To compute the FROC curve, for each threshold $T$, we plotted the average true-positive rate (TPR) per image (the ratio of the number of lesions correctly predicted in the image) versus the average number of false positives per image. In this FROC analysis, a lesion was deemed to have been correctly predicted if there was a candidate within $1.5$cm of the center-of-mass of that lesion, based on the average size of screen detected cancers.

## 3   Results

The resulting FROC curves are presented in Figure 2. We obtained a maximum sensitivity of $0.97$ with a FP per image of $3.56$ at a threshold of $0.5$ (which is the lowest threshold where the sensitivity did not further improve) using the mask R-CNN model trained with a ResNeXt-152 model. The sensitivity for different trained architectures at different average number of false positives per image are presented in Table 2.

Figure 3 shows two examples of correctly detected malignant regions for the same breast at two different views (MLO and CC, respectively).

Table 2: Sensitivity of the trained models at different average number of false positives per image. For comparison the u-net which was evaluated on the same test set from [8] is added. The u-net is not able to reach lower sensitivities than $0.15$.

| FP/image | 0.01 | 0.02 | 0.1 | maximum sensitivity |
|---|---|---|---|---|
| FasterRCNN R-101 | 0.3149 | 0.4412 | 0.6972 | 0.7889 @ 0.3223 FP/image |
| FasterRCNN X-101 (32x8) | 0.3114 | 0.4533 | 0.7301 | 0.8547 @ 0.5438 FP/image |
| FasterRCNN X-101 (64x4) | 0.3321 | 0.4412 | 0.7249 | 0.8702 @ 0.5608 FP/image |
| MaskRCNN X-101 (32x8) | 0.3183 | 0.4221 | 0.6851 | 0.9256 @ 1.6529 FP/image |
| MaskRCNN X-101 (64x4) | 0.3408 | 0.4394 | 0.6869 | 0.9429 @ 1.8302 FP/image |
| MaskRCNN X-152 (32x8) + Aug. | 0.3339 | 0.4412 | 0.6903 | 0.9689 @ 3.5602 FP/image |
| U-net [8] | – | – | – | 0.9236 @ 7.6379 FP/image |

## 4   Discussion and conclusion

In this paper, we evaluated deep learning based object detection models to localize soft tissue lesions in digital mammography. The evaluating models showed promising results in replacing the current two stage detection systems for automated detection and diagnosis. For this we evaluated both the faster R-CNN and the mask R-CNN models at different false positive levels. At an image level, we achieved a maximum sensitivity of 0.97 at 3.56 false positives per image was achieved. The best model achieved a sensitivity of 0.73, 0.45, 0.31 at 0.1, 0.02 and 0.01 FP per image, respectively. In comparison, the best proposed network by [20] achieved a sensitivity of approximately 0.93 at a false positive rate of 4 FP/image. An object detection CNN, such as the ones proposed here can reach similar performance, but without the need to introduce manual features. In this study we did not study if the performance of the CNNs was dependent on the type of DM image. In principle, introducing a

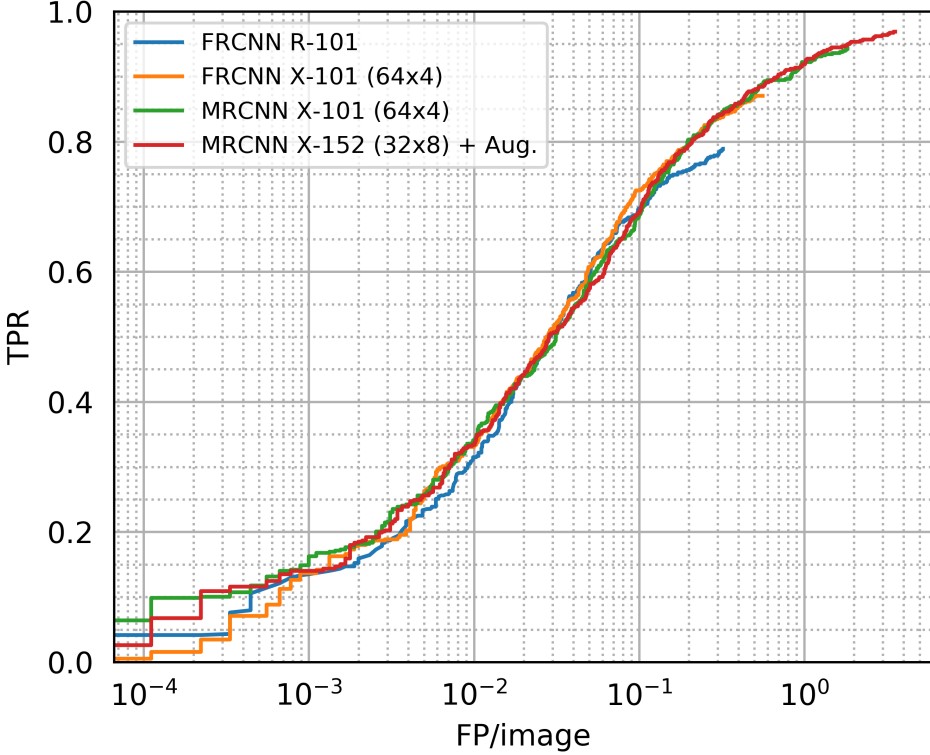

Figure 2: FROC of the best networks for each of the backends.

balanced training set across vendors and using normalization techniques might be sufficient to avoid differences between DM image qualities

Our study had some limitations. First, biopsy-verified benign soft tissue lesions were not included. These lesions might have an effect on the false positive rate when classifying between malignant and benign findings. Introducing prior DM exams, e.g. temporal information, would likely improve the performance of the network. Moreover, we only studied soft tissue lesions. Exploring similar CNN approaches to detect and classify calcifications is a topic of future work. Similarly, studying other possible classification tasks, such as differentiating between benign and malignant findings that would lead to biopsy or determining cancer aggressiveness would be of interest, especially in a screening scenario.

We noted as in [17] that adding the additional task of segmenting the lesion improves generalization. However, this requires extensive lesion-level annotations, where many datasets such as the publicly available Digital Database for Screening Mammography (DDSM) dataset and the OPTIMAM Mammography Image Database only provide bounding box annotations. A recent result in object detection [17] has shown that adding such data further improves generalization.

Another topic of future work is to study whether this two-dimensional detection model can be applied in digital breast tomosynthesis (slice-by-slice basis) or synthetic mammography images, both of which are often combined with DM as a breast cancer screening protocol. Having a robust candidate detection for all of these three types of modalities would be beneficial as it allows development of advanced computer systems that can correlate information across modalities.

In conclusion, our results encourage the use of object detection convolutional neural networks to detect and classify malignant soft tissue lesions in mammography. They could be an alternative to traditional two-step computer aided detection algorithms, which do not use the full range of information available.

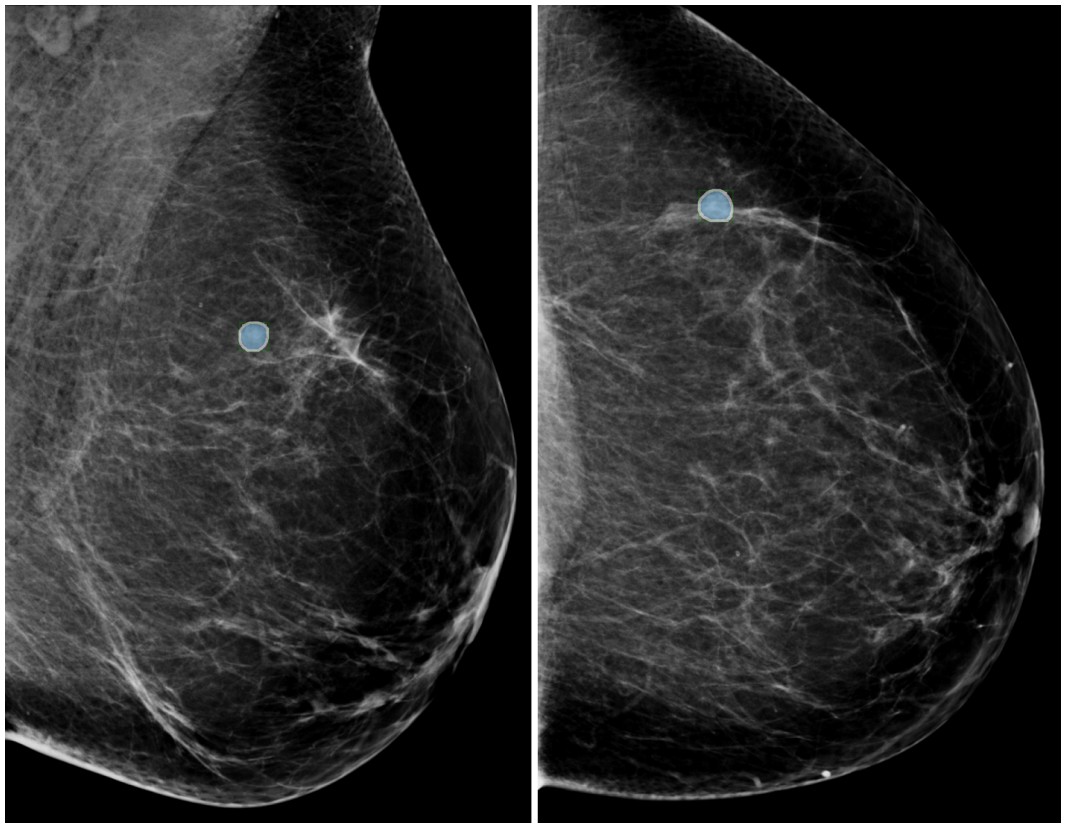

Figure 3: Examples of two small lesions correctly detected in the same breast from two different views (MLO and CC, respectively).

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
