# OpenReview forum: "Soft tissue lesion detection in mammography using deep neural networks for object detection"
_MIDL.amsterdam/2018/Conference — Submitted to MIDL 2018_

### Review · AnonReviewer1 · 2018-05-09
**A mediocre paper**

**Rating:** 2
**Confidence:** 3

**Review:**

Overall:
The paper proposes a methodology for finding breast cancer in digital mammography scans using deep learning. The problem is of high clinical importance, however, the paper presents neither any novelty in terms of an architecture nor convincing results. Additionally, the paper is written in a hard to read fashion (lengthy introduction and discussion sections).

Strengths:
+ The considered problem is interesting and practically significant.
+ The proposed approach (R-CNN) has a high potential in the medical imaging domain.

Remarks:
* Major
- For the deepest model (ResNeXt-152), the data augmentation was used also during testing. I am aware that such approach is sometimes used, however, it could introduce a big bias into the final evaluation. I would be willing to accept this result, however, I do not see a result for ResNeXt-152 without the augmentation during testing.
- The experiment is definitely not conclusive. If we remove a shaky performance of ResNeXt-152 due to additional data augmentation during testing, the next best performing model achieved slightly higher score (0.94 @ 1.83) than the current baseline (0.93 @ 4).
- The paper is a composition of well-known methods and the proposed methodology is straightforward. The only value of the paper is the experiment, however, it is not fully convincing.

* Minor
- The introduction (Section 1) is quite lengthy.
- The paper is written in a quite chaotic manner. Section 1 and Section 4 are very lengthy.

**Special Issue:**

No

---

### Review · AnonReviewer2 · 2018-05-10
**A state-of-the-art deep learning technology applied to a relevant application**

**Rating:** 3
**Confidence:** 2

**Review:**

This paper evaluates the use of the mask R-CNN technique to the application of lesion detection in mammography. The mask R-CNN technique is a recently presented technique for object instance segmentation with superior performance for object recognition. The evaluation of such an outstanding technique from the computer vision community on such an important medical imaging application is a great contribution to the field.
The paper is well structured and relatively easy to read. The previous work section and the explanation of the mask R-CNN method can be improved though. The architectural differences between the mask R-CNN and its predecessors are explained, but it’s difficult to understand for the reader why the mask R-CNN is working better than previous methods and for what kind of applications we expect to see this improvement. Also, it’s not explained why this method is expected to perform better than the two deep learning based methods for mammography lesion detection that are mentioned in the paper (references [8] and [20]). Furthermore, the authors modified the original mask R-CNN (and faster R-CNN) by adding the multi-scale feature pyramid network architecture. This is not motivated and the method is not compared against the original single-scale architecture. The authors should put more effort in trying to educate the community why the new method (mask R-CNN with feature pyramid networks) is working so well and which kind of applications (with what kind of training data) could also potentially benefit from it.
Another point of improvement of the paper is the evaluation. Specifically, the recent paper on the same application (reference [20]; co-authored by some of the authors of this paper) does not seem to be directly compared against the proposed method. The performance of the previous method is mentioned, but it seems to be on a different data set. Also, for one of the variants of the proposed method the authors apply augmentation at test time. It’s unclear why that is done and if the evaluation is still fair.

**Special Issue:**

No

---

### Review · AnonReviewer3 · 2018-05-14
**Out-of-the-box deep learning based object detection applied to clinically relevant problem**

**Rating:** 2
**Confidence:** 2

**Review:**

The paper discusses the performance of state-of-the-art object detection algorithms (Faster R-CNN and Mask R-CNN) from the computer vision field on the task of soft-tissue lesion detection in mammography images. This is a clinically relevant application where accurate computer-aided detection methods could make a real impact in improving current screening procedures. The methods are pretty much applied out-of-the-box and therefore the paper lacks technical novelty. Still, a comprehensive evaluation of these methods on this clinically relevant task could be an important contribution for the medical imaging community. However, there are several problems with the evaluation and comparison to previous methods.

1)	For a clinically well defined task such as this, I would expect a more comprehensive comparison to the state-of-the-art and previous work. The comparison to Ref. 20 seems to be just a quotation of the results from their paper. A direct comparison on the same data to other well established two-stage approaches for lesion detection would be desired. This would also strengthen the fundamental claim of this paper that a direct approach is better than the more traditional two-stage cascaded approaches.
2)	Data augmentation is used during testing for one of the methods but it is not explained how exactly this is being done. Furthermore, the evaluation of the same method without augmentation is missing, making it hard to quantify its contribution.


Minor comments:
1)	Caption of Table 2, the sentence “The u-net is not able to reach lower sensitivities than 0:15.” should refer to false-positive rates, I presume. Furthermore, a more detailed description about this method should be added.

2)	It would be good to add some evaluation of the causes for false-positives or at least to show some examples.


**Special Issue:**

No

---

### Decision · Program_Chairs · 2018-05-15
**Paper101 Acceptance Decision**

Reject